# MicroRNA-7: A New Intervention Target for Inflammation and Related Diseases

**DOI:** 10.3390/biom13081185

**Published:** 2023-07-28

**Authors:** Chao Chen, Mengmeng Guo, Xu Zhao, Juanjuan Zhao, Longqing Chen, Zhixu He, Lin Xu, Yan Zha

**Affiliations:** 1School of Medicine, Guizhou University, Guiyang 550025, China; 2Department of Immunology, Zunyi Medical University, Zunyi 563000, China; 3Specifc Key Laboratory of Gene Detection and Treatment of Guizhou Province, Zunyi 563000, China; 4Collaborative Innovation Center of Tissue Damage Repair and Regeneration Medicine of Zunyi Medical University, Zunyi 563000, China

**Keywords:** microRNA-7, inflammation, immune response, T cells, intervention

## Abstract

MicroRNAs (miRNAs) are a class of small noncoding RNA that can regulate physiological and pathological processes through post-transcriptional regulatory gene expression. As an important member of the miRNAs family, microRNA-7 (miR-7) was first discovered in 2001 to play an important regulatory role in tissue and organ development. Studies have shown that miR-7 participates in various tissue and organ development processes, tumorigenesis, aging, and other processes by regulating different target molecules. Notably, a series of recent studies have determined that miR-7 plays a key regulatory role in the occurrence of inflammation and related diseases. In particular, miR-7 can affect the immune response of the body by influencing T cell activation, macrophage function, dendritic cell (DC) maturation, inflammatory body activation, and other mechanisms, which has important potential application value in the intervention of related diseases. This article reviews the current regulatory role of miR-7 in inflammation and related diseases, including viral infection, autoimmune hepatitis, inflammatory bowel disease, and encephalitis. It expounds on the molecular mechanism by which miR-7 regulates the occurrence of inflammatory diseases. Finally, the existing problems and future development directions of miR-7-based intervention on inflammation and related diseases are discussed to provide new references and help strengthen the understanding of the pathogenesis of inflammation and related diseases, as well as the development of new strategies for clinical intervention.

## 1. Introduction

Under normal circumstances, inflammation is a physiological defense response of the body to stimulation, involving a complex and fine-tuned regulatory process that is conducive to eliminating pathogens and promoting tissue repair [1]. It is now known that the inflammatory process is closely related to the body’s innate and adaptive immune responses, involving the activation and function of innate immune cells, such as macrophages and DCs, and adaptive immune cells, such as T cells and B cells. However, if the immune response process is abnormal, the continuous development of inflammation can lead to autoimmune or inflammatory diseases, neurodegenerative diseases, and even cancer [2]. The inflammatory response involves innate and adaptive immune response processes mediated by immune cells with the participation of histiocytes and molecules. Therefore, the regulatory mechanisms involved in the development of inflammation and related diseases are very complex. The analysis of these mechanisms is of great significance for understanding the mechanisms of related diseases and clinical treatment.

MicroRNAs (miRNAs, miRs) are noncoding single-stranded RNA molecules encoded by endogenous genes with a length of about 22 nucleotides. By specifically pairing with the 3’-untranslated region (UTR) base of the target mRNA, miRNAs can inhibit translation or induce target mRNA degradation [3], thereby regulating gene expression after transcription to support precise regulation of cell growth, proliferation, development, apoptosis, and stress response [4,5]. A series of studies have confirmed the important regulatory role of miRNA molecules in the occurrence of inflammation and related diseases. For example, the NF-κB-miR-155 axis operates in conjunction with the NF-κB-miR-146a axis to regulate the intensity and duration of inflammation. Mice lacking miR-146a develop an inflammatory syndrome, autoimmunity, and cancer. Interestingly, miR-146a-deficient cells exhibit increased expression of miR-155, and their pro-inflammatory phenotype may be diminished by the depletion of miR-155 [6]. DC activation can be enhanced by miR-223 and the subsequent differentiation of naive T cells into Th1, Tregs, and Th17 effector cells, thus promoting the process of experimental autoimmune encephalomyelitis (EAE) [7]. However, considering the complexity of miRNA family members, systematic exploration of the role of specific miRNA molecules in the progression of inflammation is of great value for exploring the regulatory mechanisms of key signaling pathways in inflammation-related diseases and for formulating treatment strategies for clinically related diseases.

The evolutionarily conserved miR-7 was discovered by Lagos Quintana et al. in 2001 [8] [Figure 1]. In the human gene sequence, it was found that the same miR-7 mature sequence could result from the transcription and processing of three different DNA sequences, which were named miR-7-1 (on chromosome 9), miR-7-2 (on chromosome 15), and miR-7-3 (on chromosome 19) [9,10]. The mature miR-7 sequence is generated from three genes to ensure that miR-7 can be fully transcribed, suggesting that it plays an important regulatory role in the development of tissues and cells in the body. Most previous studies have shown that miR-7 is a tumor suppressor, and the abnormal expression of miR-7 is usually closely related to the development of tumors [11,12]. Our previous studies also successively found that miR-7 regulates the growth and metastasis of human lung cancer cells [13,14,15,16]. Interestingly, in recent years, the regulation of miR-7 by inflammation has gradually attracted attention. This review summarizes how miR-7 plays a key role in inflammation-related diseases and highlights its potential role as a new inflammatory target.

## 2. Regulatory Mechanism of miR-7 Expression

The expression of miR-7 differs between cell types and tissue environments and is coordinated and regulated by multiple pathways to respond to cell signals [Figure 2].

### 2.1. The DNA Methylation

Methylation is an important process that modifies proteins and nucleic acids and regulates gene expression and shutdown [17]. Many studies have confirmed that epigenetic changes through DNA methylation can reduce the expression of some miRNAs. For example, chronic obstructive pulmonary disease (COPD) is an inflammatory lung disease. A recent study showed increased methylation of miR-7 in the alveolar epithelial cells of COPD patients [18]. Since miR-7 has a suppressive role in lung cancer, its altered DNA methylation may help explain the high incidence of lung cancer in these patients.

### 2.2. circRNAs

Circular RNAs (circRNAs) are a novel class of noncoding RNA molecules that widely exist in the cytoplasm of eukaryotic cells. Studies have found that cerebellar-degeneration-related protein 1 antisense RNA (CDR1as, also referred to as ciRS-7) can regulate the activity of miR-7 and exert specific biological effects. Structurally, ciRS-7 contains approximately ~70 anti-miR-7 sequences in tandem and acts as an endogenous, anti-complementary miR-7 “sponge” that adsorbs, binds, and, hence, quenches natural miRNA-7 functions [19,20]. In inflammation-associated sporadic Alzheimer’s disease (AD), ciRS-7 expression is downregulated, leading to excess ambient miR-7 that appears to drive selective downregulation of miR-7-sensitive mRNA targets, such as that encoding the ubiquitin-conjugating enzyme E2A (UBE2A). UBE2A normally acts as a central effector in the ubiquitin-26s proteasome system, coordinating the clearance of amyloid peptides by proteolysis. Thus, dysfunction or defect in the ubiquitin-26s proteasome system could contribute to amyloid accumulation and the formation of senile plaque deposits [21].

### 2.3. lncRNA

Emerging evidence shows that a variety of long noncoding RNA (lncRNA) can play a vital role in various cells and diseases by regulating the expression of miR-7. H_2_O_2_ can induce the body to produce an oxidative stress response, forming many free radicals, leading to inflammation, decreased cell vitality, and other damage. Zhao et al. found that lncRNA antisense noncoding RNA in the INK4 locus (ANRIL) can enhance the phosphorylation levels of the key kinases in the mTOR and MEK/ERK pathways by downregulating miR-7, thereby significantly attenuating the oxidative damage of human trabecular meshwork (TM) cells induced by H_2_O_2_ stimulation [22]. Inflammatory processes can change the structure and function of diabetic retinopathy (DR). LncRNA zinc finger antisense-1 (ZFAS1) is upregulated in human retinal endothelial cells (hRECs) cultured with high glucose. Further studies found that ZFAS1 may positively promote endothelial ferroptosis in DR via competitive binding with miR-7-5p and regulating the expression of its downstream molecule, acyl-CoA synthetase long-chain family member 4 (ACSL4) [23]. In the Parkinson’s disease (PD) model, the expression of long intergenic noncoding RNA 00943 (LINC00943) was increased in N-methyl-4-phenylpyridine (MPP+)-inducted SK-N-SH cells. Silencing LINC00943 could promote the viability of SK-N-SH cells treated with MPP+ and inhibit cell apoptosis and the inflammation response. Mechanistic analysis showed that LINC00943 acts as a sponge for miR-7-5p and regulates the expression of chemokine (C-X-C motif) ligand 12 (CXCL12) [24]. Small nucleolar RNA host gene 1 (SNHG1), as a competing endogenous long-chain noncoding RNA of miR-7, regulates the expression of node-like receptor protein 3 (NLRP3), leading to the activation of the NLRP3 inflammasome in PD [25].

### 2.4. Transcription Factors (TFs)

Aging is closely related to inflammation, and studies have found that miR-7 transcription is upregulated in senescent cells. Midgley et al. investigated the reasons for the upregulation of miR-7 transcription in senescent fibroblasts. They found multiple STAT-binding sites on the miR-7 promoter, including STAT1, the IFN-stimulating regulatory element (ISRE), and the IFN-stimulating transcription factor 3-γ (ISGF3G). These potential TFs binding sites strongly suggest that STAT activation may promote the upregulation of miR-7 expression [26]. Recently, our research group found that the precursor sequence miR-7b is the main source of mature miR-7 expression in CD4^+^T cell activation in a mouse autoimmune hepatitis model. In addition, transcription factor C/EBPα can positively regulate the expression of miR-7 by directly binding to the promoter region of the miR-7b gene [27]. These results reveal that lineage-specific TFs may control the selective expression of miR-7.

### 2.5. Target-Directed miRNA Degradation (TDMD)

In mammals, TDMD has been shown to powerfully regulate specific miRNAs. For example, CYRANO is a highly conserved lncRNA. Its single site has extensive complementarity with miR-7, which has been demonstrated to trigger the robust attenuation of this miRNA [28]. However, the molecular mechanisms that lead to miRNA decay through this pathway remain elusive. Two research results on the new mechanism of cellular miRNA degradation published at the same time in the Science magazine show that, generally, in the process of miRNA-mediated regulation, miRNAs act jointly with Argonaute (AGO) proteins to repress target mRNAs. In contrast, Zinc finger SWIM-type containing 8 (ZSWIM8) is required for Cyrano-directed miR-7 degradation and other known examples of TDMD. Target sites paired with the miR-7 3′ region induce conformational changes in ZSWIM8 ubiquitin ligase recognition. This ligase polyubiquitinates AGO, leading to proteasomal degradation of miR-7-containing complexes. The release of miR-7 can lead to its further degradation by cytoplasmic RNases [29,30].

### 2.6. Other Factors

miR-7-1 is an intron of the heterogeneous nuclear ribonucleoprotein K (HNRNPK) gene [31,32,33], and miR-7-3 is an intron of PGSF1a [10,33]. As an intronic miRNA molecule, the expression level of miR-7 is also affected by the self-expression regulatory mechanisms of host genes. For example, when the expression of HNRNPK and PGSF1a increases or decreases under certain physiological conditions and diseases of the host, the expression levels of pri-miR-7-1 and pri-miR-7-3 will also change accordingly. Thus, the level of mature miR-7 will be affected.

Small interfering RNA (siRNA) targeting poliovirus (PV) and other viruses can effectively inhibit virus replication and has been developed as an antiviral nucleic acid drug. For example, Zhang et al. found a specific siRNA-100 targeting the region between nucleotides 100–125 of the 5’-UTR of PV, which plays a key role in inhibiting PV replication while leading to the upregulation of host miR-7 [34]. However, the specific mechanism needs to be further clarified.

Hydroxycholesterol (HC), an inhibitor of sterol regulatory element binding protein (SREBP) signaling and liver X receptor (LXR) agonist and peroxisome proliferator activated receptor-α (PPAR-α) antagonist treatment, downregulated miR-7 expression levels. Regulation by two different metabolic inhibitors highlights a potential role for miR-7 in hepatic lipid pathways. qRT-PCR analysis and miRNA microarray data further confirmed that miR-7-2 is the genomic locus where PPAR-α regulation occurs [35].

## 3. miRNA-7 and Inflammatory Diseases

miR-7 has different regulatory mechanisms in different inflammation-related diseases and transmits cellular signals through multiple pathways [Table 1] [Figure 3]. Due to this, a detailed understanding of the regulatory mechanisms of miR-7 may contribute to the development of new miR-7-based therapeutics for treating human inflammatory-related diseases.

**Table 1 biomolecules-13-01185-t001:** Regulatory mechanisms of miR-7 in different inflammatory-related diseases.

Inflammatory Related Diseases	Genes/Drugs That Regulate miR-7	Target Genes of miR-7	Immune Cells or Molecules	Outcome	Expression Level of miR-7	Remark	Ref.
Lung cancer	HuR	PI3K	unknown	Inhibit the growth and metastasis of lung cancer cells by inhibiting the TLR9 signaling pathway	(↓) 95D cell	miR-7 mimic transfected (in vitro) plasmids transfected (in vivo)	[14,15]
Aging	STAT1	EGFR	TGF-β1, IFN-γ	Impair age-associated loss of EGFR and hyaluronan-dependent differentiation in fibroblasts.	(↑) in aged fibroblasts	miR-7 mimic/locked nucleic acids (LNA) targeting miR-7 transfected (in vitro)	[26,36]
AIH	C/EBPα	MAPK4	CD4^+^T, IFN-γ, IL-4	CD4^+^T cells with miR-7 deficiency exacerbate the pathology of AIH	(↑) in liver tissue in mice	plasmids transfected (in vitro)	[27]
NAFLD	PPAR-α	ERLIN2, NR1H2	unknown	Promote hepatocellular lipid accumulation	(↑) in fatty liver in mice	miR-7 mimic transfected (in vitro)	[35]
		LAMP1	IL-1β	Promote microvascular endothelial hyperpermeability	(↑) in the sEVs of fatty liver	hepatic small extracellular vesicles (sEVs)	[37]
		YY1	IL-1β, IL-6, TNF-α, IFN-γ	Improve hepatic steatosis and steatohepatitis	(↓) Nonalcoholic fatty liver in zebrafish in mice	miR-7 mimic transfected (in vitro)	[38]
PD	LINC00943	CXCL12	IL-1β, TNF-α	Promote cell viability, repress apoptosis and the inflammatory response	(↓) in MPP+-inducted SK-N-SH cells	miR-7 mimic/inhibitor transfected (in vitro)	[24]
	SNHG1	NLRP3	Microglial, IL-6, TNF-α, IL-1β, IL-18	Inhibit the activation of NLRP3 inflammatories	(↓) in LPS-induced BV2 cells	miR-7 mimic/inhibitor transfected (in vitro)	[25]
		NLRP3	Microglial, IL-1β	Inhibit the activation of NLRP3 inflammasome and attenuate dopaminergic neuronal degeneration.	(↓) in the serum samples of PD patients	miR-7 mimic transfected (in vitro) stereotactically injected miR-7 mimics (in vivo)	[39,40]
		RelA	TNF-α	Protect neurons, prevent cell death and promote glycolysis.	(↓) in MPP(+) -inducted SH-SY5Y cells	plasmids transfected (in vitro)	[41,42]
		KLF4	unknown	Inhibit neuronal cells apoptosis	(↓) in MPP(+) -inducted SH-SY5Y cells	miR-7 mimic transfected (in vitro)	[43]
		Bax, Sirt2	unknown	Inhibit neuronal cells apoptosis	(↓) in MPP(+) -inducted SH-SY5Y cells	miR-7 mimic transfected (in vitro)	[44]
	Crocin	α-syn	unknown	miR-7 has a neuroprotective effect	(↓) in striatal tissue in mice	Unknown	[45]
		α-syn	unknown	Protect neuronal cells from oxidative stress	(↓) in the MPTP-induced neurotoxin model of PD in cultured cells and in mice	plasmids transfected (in vitro)	[46]
		BDNF	unknown	a strong self-protective mechanism in rats at the early stage of PD.	(↓) in peripheral blood of rats with Atrazine-induced PD	Unknown	[47]
		SNCA	unknown	Protect neuronal cells	(↓) in the substantia nigra of PD patients	miR-7 lentiviral vector transfected (in vitro) stereotactically injected miR-7 lentiviral vector (in vivo)	[48]
Neuro- inflammation		TLR4	Microglial, TLR4, IL-1β, TNF-α, IL-8	Alleviate the inflammatory response	(↓) in patients with intracerebral hemorrhage	miR-7 mimic/inhibitor transfected (in vitro) mixture of the rAAV9-ZsGreen-miR-7 virus transfection agent (adenovirus vector)were injected into the hematoma region of the rat (in vivo)	[49]
	Nicorandil	Herpud2	IL-1β, TNF-α	Reduce the inflammatory response and astrocyte damage	(↓) in oxygen-glucose deprivation inducted astrocytes	miR-7 mimic transfected (in vitro)	[50]
		RORα	Microglial, IL-1β, TNF-α, IL-6, TGF-β	Negatively control the pathology of BTI	(↑) in brain tissue in BTI mice	miR-7 mimic/inhibitor transfected (in vitro)	[51]
IBD	IL-1β	CD98	IL-1β	Improve the intestinal inflammation	(↓) in inflamed colon tissues of crohn patients	plasmids transfected (in vitro)	[52]
	Infliximab	RNF183	IL-1β, TNF-α, IL-6, IL-8	Alleviate the process of IBD	(↓) in inflamed colon tissues of IBD patients and colitic mice	miR-7 mimic/inhibitor transfected (in vitro)	[53]
		TFF3	unknown	Promote the process of IBD	(↑) in the lesional tissue of IBD patients	miR-7 mimic/inhibitor transfected (in vitro)	[54,55]
	C/EBPα	EGFR	IL-6, IL-10, TNF-α, TGF-β, CD4^+^T, B, CD8^+^T	Promote the process of IBD	(↑) in colonic IECs in colitis of IBD patients	miR-7 mimic/inhibitor transfected (in vitro) IEC-specific miR-7 silencing expression vector tail vein injected (in vivo)	[56]
Sepsis		Bad, Bax, Caspase-3, BCL-2	T	Inhibit T lymphocyte apoptosis and reduce the mortality of sepsis model mice	(↑) in plasma circulating exosomes in sepsis patients	plasmids/adenovirus transfected (in vitro) Exosomes(in vivo)	[57]
	circVMA21	PPAR-α	unknown	Promote the process of sepsis-engendered AKI	(↑) in sepsis patients’ serums and LPS-stimulated HK2 cells	miR-7 mimic transfected (in vitro)	[58]
SLE		PTEN	B, Tfh, IL-21	Promotes disease manifestations in MRL^lpr/lpr^ lupus mice, and abnormal splenic B cell subtypes	(↑) in B cells in SLE patients	miR-7 antagomir transfected (in vitro/vivo)	[59,60]
OA	ciRS-7	PI3K	IL-1β, IL-17A	Promote cartilage degradation and autophagy defects	(↑) in blood samples from OA patients	miR-7 mimic/siRNA transfected (in vitro)	[61,62]
		MEGF9	IL-1β	Exacerbate the OA process.	(↑) in blood samples from OA patients	miR-7 mimic/siRNA transfected (in vitro) injected miR-7 lentiviral vector (in vivo)	[63]
	SNHG15	KLF4	IL-1β	Promote OA progression	(↑) in human OA knee cartilage tissues	miR-7 mimic/inhibitor transfected (in vitro)	[64]
		SEMA6D	unknown	Promote OA progression	(↑) in IL-1β-induced osteoarthritic C28/I2 chondrocytes	miR-7 mimic/inhibitor transfected (in vitro/vivo)	[65]
Gastric tumors	macrophage derived factors	unknown	IL-1β, TNF-α	Inflammation-induced repression of miR-7 in gastric tumor cells	(↓) in human gastric cancer tissue	miR-7 mimic transfected (in vitro)	[66]
Liver cancer	HBx	Maspin	unknown	Promotes HCC progression	(↑) in HBx-expressing HCC cells	miR-7 inhibitor transfected (in vitro)	[67]
	HBx	EGFR	unknown	Inhibit HCC progression	(↑) in HBx-expressing HCC cells	miR-7 inhibitor transfected (in vitro)	[68]
ALI		KLF4	IL-1β, TNF-α, IFN-γ, CD4^+^T DC, CD8^+^T	Promote the lung pathology of ALI mice.	(↑) in the lung tissues of LPS-induced mouse ALI models	Unknown	[69]
GDM		RAF1, IRS1/2	IL-6R	Cause chronic low-grade inflammation and promote the progression of GDM.	(↑) in maternal blood of GDM patients	Unknown	[70]
RV		NSP5	unknown	Inhibit RV replication.	(↑) in RV-infected cells	miR-7 mimic/inhibitor transfected (in vitro) intragastric inoculation of miR-7 agomir/antagomir (in vivo)	[71]
CAG		TFF2	unknown	Inhibit the development and progression of SPEM.	(↓) in human SPEM lesions/gastric cancer tissue	Unknown	[72]
HIV		unknown	IL-6	miR-7 was significantly upregulated in HIV-1 infection after treatment.	(↑) in PBMCs from HIV-1-infected individuals	Unknown	[73]
Immuno- inflammatory	TLR4	FAM177A	Macrophage, TLR4, IL-1β, TNF-α, IL-6, IL-12	Negatively regulate TLR4 signaling pathway.	(↑) in TLR4 signalling-activated bone marrow-derived macrophages stimulated by LPS	miR-7 mimic transfected (in vitro)	[74]

(↑): High expression (↓): Low expression

### 3.1. Hepatitis

#### 3.1.1. Autoimmune Hepatitis (AIH)

Autoimmune hepatitis is a chronic, progressive liver inflammatory disease mediated by an autoimmune response and closely related to the activation and infiltration of immune cells. CD4^+^T cells are important cellular components of the immune response function, mediating the occurrence and development of AIH [75]. In a recent study, our research group found that miR-7 deficiency can lead to pathological deterioration in a ConA-induced murine acute autoimmune liver injury model, accompanied by a hyperactivated state of CD4^+^T cells [27]. Further studies found that in CD4^+^T cells, the transcription factor C/EBPα can positively regulate the expression of miR-7 by directly binding to the core promoter region of the miR-7b gene. In addition, global gene analysis revealed that mitogen-activated protein kinase 4 (MAPK4) is a target of miR-7, and the loss of MAPK4 could ameliorate the activation status of CD4^+^T cells [27]. Our evidence shows that the C/EBPα/miR-7 axis negatively regulates the activation and function of CD4^+^T cells through MAPK4 and is a new target for clinical intervention in AIH. However, whether this axis can regulate the involvement of other cells in AIH remains to be studied.

#### 3.1.2. Nonalcoholic Fatty Liver Disease (NAFLD)

NAFLD is the most common chronic liver disease. During the development of NAFLD, small extracellular vesicles (sEVs) derived from damaged hepatocytes induce inflammation and fibrogenesis [76]. The most recent findings show that elevated lipid levels in the liver induce the secretion of hepatic sEVs enriched in miR-7, and miR-7 is the most significantly expressed miRNA in hepatic sEVs. It inhibits lysosomal associated membrane protein 1 (LAMP1) transcription by directly binding to the 3’-UTR of LAMP1 mRNA. Ultimately, the hyperpermeability of the microvascular endothelium is promoted through the LAMP1/CathepsinB/NLRP3 inflammasome axis. Inhibition of miR-7 improved the integrity of the microvascular endothelial barrier [37]. The accumulation of lipids in hepatocytes can trigger chronic inflammatory responses. Studies have found that miR-7 is a PPAR-α regulated miRNA that targets ERLIN2 and NR1H2 to activate SREBP signaling and promote hepatocyte lipid accumulation [35]. In another study, Lai et al. applied miRNA-sponge (miR-SP) technology to destroy the activity of liver miR-7a and induce early NAFLD and nonalcoholic steatohepatitis (NASH) in zebrafish [38]. Regarding the mechanism, Yin Yang 1 (YY1) was found to be a new target of miR-7a, and it was concluded that miR-7a-SP-stabilized YY1 is beneficial for lipid anabolism. Meanwhile, the expression of inflammation-related genes [Interleukin 1 beta (IL-1β), IL-6, Tumor necrosis factor alpha (TNF-α), Interferon gamma (IFN-γ), Nuclear factor kappa B (NF-κB) and NF-κB2] was increased. However, increasing the expression of miR-7a can improve hepatic steatosis and steatohepatitis in the miR-7a-SP model.

### 3.2. Parkinson’s Disease (PD)

PD is a common neurodegenerative disease. Neuroinflammation plays an important role in the pathogenesis of PD. SNHG1 expression was increased and miR-7 expression was decreased in PD patients and in vitro cell models. Interestingly, downregulation of SNHG1 in vitro increased miR-7 levels and inhibited lipopolysaccharide (LPS)-induced microglia activation and inflammation, thereby preventing the potential loss of dopaminergic neurons in the substantia nigra. Mechanistically, miR-7, as a regulator of the SNHG1/NLRP3 axis, plays an important regulatory role in the inflammatory effects of microglia [25]. The study found that the NLRP3 inflammasome was activated in the serum of PD patients and in a murine PD model. Zhou et al. demonstrated for the first time that NLRP3 is a target gene of miR-7. Stereotaxic injection of miR-7 mimics in the striatum of mice significantly inhibited the activation of the NLRP3 inflammasome in microglia, thereby attenuating dopaminergic neuronal degeneration [39]. Another similar study found that stereotaxic injection of miR-7 mimics into the lateral ventricle significantly inhibited NLRP3 inflammasome activation and improved neurogenesis in mice’s subventricular zone (SVZ) [40]. Choi et al. confirmed that miR-7 protects neurons by targeting and inhibiting the expression of RelA and alleviating the inhibition of NF-κB [41]. In addition, miR-7 can also prevent 1-methyl-4-phenylpyridinium [MPP(+)]-induced cell death by downregulating NF-κB p65 (RelA), increasing glucose transporter 3 (Glut3) expression, and promoting glycolysis [42]. This protective role of miR-7 could be used to correct the defect in oxidative phosphorylation in PD. Furthermore, miR-7 protected against MPP(+)-induced apoptosis in neuronal cells by directly targeting Krüppel-like factor 4 (KLF4) [43], and it also inhibited MPP(+)-induced neuronal cell apoptosis by directly targeting B-cell lymphoma-2-associated X protein (Bax) and sirtuin2(Sirt2) [44]. Crocin is a carotenoid in saffron that has beneficial effects on neurodegenerative diseases through anti-apoptotic, anti-inflammatory, and antioxidant activities. Studies have shown that crocin has a good neuroprotective effect on rotenone-induced PD by activating the phosphatidylinositol-3 kinase (PI3K)/protein kinase B (Akt)/mechanistic target of rapamycin (mTOR) axis and enhancing miR-7 and miR-221 [45].

Abnormal expression and aggregation of α-synuclein (α-syn) can induce neuroinflammation and interfere with the process of nerve repair, an important feature of PD [77]. Junn et al. found that under normal circumstances, miR-7 in neurons inhibits the level of α-syn protein through the 3’-UTR of α-syn RNA, thereby protecting cells from oxidative stress. In PD models, miR-7 expression was reduced, resulting in increased α-syn expression [46]. Li et al. found that miR-7 regulates the brain-derived neurotrophic factor (BDNF)/α-syn axis in the early stages of PD, and miR-7 regulates the expression of BDNF through an autoregulated mechanism [47]. In addition, miR-7 can also accelerate the clearance of α-syn and its aggregates by promoting autophagy in differentiated ReNcellVM cells [78]. McMillan et al. found that miR-7 downregulates the expression of α-syn by binding to the 3’UTR of the synuclein alpha non-a4 component of amyloid precursor (SNCA) gene and inhibiting its translation, thereby protecting neuronal cells [48]. Further studies also demonstrated that the over-expression of miR-7 by AAV vectors in the mouse brain could inhibit the formation and proliferation of pathological α-syn and protect α-syn prefabricated fibril-induced neurodegeneration and behavioral deficits [79]. All these findings suggest that miR-7 has the potential to slow down the progression of PD.

### 3.3. Neuroinflammation

It is well known that inflammation is a common pathological basis for various neurological diseases. Many recent studies have shown that miR-7 is involved in neuroinflammation. For example, Zhang et al. found that the expression of miR-7 in a rat cerebral hemorrhage model and a microglial inflammation model tissue was significantly lower than that in the normal control group. Over-expression of miR-7 can inhibit the expression of Toll-like receptor 4 (TLR4), thereby alleviating the inflammatory response induced by LPS in microglia [49]. Conversely, oxygen-glucose deprivation treatment can lead to significant downregulation of miR-7 expression, resulting in astrocytic inflammatory damage. However, k-ATP channel opener (nicorandil) can significantly reverse the expression level of miR-7, and miR-7 can target HERPUD family member 2 (Herpud2) 3’UTR, inhibiting the expression of endoplasmic reticulum-related proteins and thereby reducing the inflammatory response and astrocyte damage [50]. In addition, the latest evidence from our research group shows that miR-7 is upregulated in the brain tissue of LPS-induced brain tissue inflammation (BTI) mice. Interestingly, the lack of miR-7 significantly aggravated the pathology of BTI. Further research found that RAR-related orphan receptor alpha (RORα) is a new target of miR-7. Unlike previous reports that miRNAs exert biological functions opposite to their targets, the inhibition of RORα in the absence of miR-7 aggravates rather than reverses the pathology of BTI. Considering the upregulation of RORα expression in the absence of miR-7, a new network model in which miR-7 cooperates rather than antagonizes its target gene RORα to control the pathology of BTI is proposed in this study [51]. The above data further support the important role of miR-7 in neuroinflammation, which may help to explore the molecular mechanisms of the development of inflammatory brain diseases.

### 3.4. Inflammatory Bowel Disease (IBD)

Inflammatory bowel disease, which includes Crohn’s disease (CD) and ulcerative colitis (UC), refers to chronic inflammatory diseases affecting the gastrointestinal tract. The destruction of pro-inflammatory factors can lead to the occurrence of IBD. For example, IL-1β reduces the level of miR-7 in CD colon tissue, thereby increasing the expression of the miR-7 target molecule CD98, leading to aggravated intestinal inflammation [52]. Blocking TNF activity with infliximab effectively induces mucosal healing in CD patients and has become a major therapeutic tool for IBD, but the exact mechanism has not been fully elucidated [80]. Yu et al. found in their study that infliximab treatment led to increased expression of miR-7 in inflamed mucosa. Furthermore, miR-7 could reduce the ubiquitination and degradation of NF-κB inhibitor alpha (IκBα) by negatively regulating RING finger protein 183 (RNF183), thereby inhibiting the NF-κB pathway and alleviating the process of IBD [53].

Contrary to the results of several studies mentioned above, miR-7 also plays a harmful role in the process of IBD. For instance, Trefoil factor 3 (TFF3) plays an important role in intestinal mucosal injury and healing, contributing to the treatment of IBD. Guo et al. found that miR-7-5p expression was increased in the diseased tissues of IBD patients, and miR-7-5p could directly target the 3’UTR of TFF3 mRNA [54]. In terms of mechanism, over-expression of miR-7-5p can inhibit the PI3K/Akt signaling pathway by targeting TFF3. Furthermore, the PI3K/Akt signaling pathway can produce a feedback regulation effect on miR-7-5p, inhibit the activity of this signaling pathway, increase the expression level of miR-7-5p, and further enhance the inhibition of miR-7-5p in cell proliferation and migration [55]. Recently, our group also found that miR-7 was significantly up-regulated in colonic intestinal epithelial cells (IECs) in colitis patients. It was further demonstrated that mature miR-7, mainly derived from C/EBPα-operated miR-7a-1, controls the proliferation and inflammatory cytokine secretion of IECs through the epidermal growth factor receptor (EGFR)/NF-κB/Akt/Erk pathway. Finally, IEC-specific miR-7 silencing promoted the proliferation and transduction of the NF-κB pathway in IECs, reducing the pathological damage of colitis [56].

### 3.5. Sepsis

Sepsis is a systemic inflammatory response syndrome caused by bacteria and other pathogenic microorganisms invading the body. Immunosuppression caused by T lymphocyte apoptosis is an important pathological feature and a problem in sepsis. Using next-generation sequencing (NGS), Deng et al. found that the expression of hsa-miR-7-5p in the circulating exosomes derived from the plasma of patients with sepsis (Sepsis-Exos) was significantly increased. Hsa-miR-7-5p can downregulate the BCL2-associated agonist of cell death (Bad), active Caspase-3, and Bax while upregulate the anti-apoptotic gene Bcl-2, thereby inhibiting LPS-induced T lymphocyte apoptosis in vitro. In addition, it has been confirmed in vivo that Sepsis-Exos can inhibit T lymphocyte apoptosis and reduce the mortality of sepsis model mice [57]. Another study found that the circRNA vacuolar ATPase assembly factor VMA21 (circVMA21) was reduced in the serum of sepsis patients and LPS-stimulated HK2 cells. circVMA21 was identified as the sponge for miR-7-5p, and over-expression of circVMA21 could have an inhibitory effect on miR-7-5p through targeting PPAR-α. miR-7-5p inhibition could reverse the suppressive effect on cell viability and the promotional effects on cell apoptosis, inflammation, and oxidative stress in HK2 cells mediated by LPS [58]. These studies demonstrated the important regulatory role of miR-7 in the occurrence of sepsis. However, whether the differences in miR-7 function are related to different cell subsets remains to be investigated.

### 3.6. Systemic Lupus Erythematosus (SLE)

SLE is an autoimmune disease characterized by B cell hyperreactivity and the production of pathogenic autoantibodies, which leads to multi-organ immune complex deposition, chronic inflammation, and organ damage [81]. Wu et al. found that the expression level of miR-7 was elevated in these B cells, and miR-7 could directly target the phosphatase and tensin homolog (PTEN) 3’-UTR, inhibit the expression of PTEN mRNA and protein, and increase the activation of the PI3K/Akt signaling pathway [59]. The team further found that miR-7-mediated downregulation of PTEN/Akt signaling in MRL^lpr/lpr^ lupus mice promoted the differentiation of B cells into plasmablasts/plasma cells and the formation of spontaneous germinal centers. However, miR-7 antagonists can effectively improve disease manifestations in MRL^lpr/lpr^ lupus mice, normalize splenic B cell subtypes, downregulate signal transducer and activator of transcription 3 (STAT3) phosphorylation and IL-21 production, and reduce Tfh expansion [60]. These data suggest that miR-7 is a potential therapeutic target for SLE.

### 3.7. Osteoarthritis (OA)

OA is a common chronic degenerative joint disease in middle-aged and older adults, characterized by articular cartilage destruction and arthritis. The research found that ciRS-7 expression decreased and miR-7 expression increased in OA blood samples, which could be further enhanced by IL-1β induction [61]. Another study found that the ciRS-7/miR-7 axis can ameliorate cartilage degradation and autophagy defects by activating PI3K/Akt/mTOR [62]. Another basic study revealed that the miR-7/EGFR/multiple EGF-like domains 9 (MEGF9) axis could regulate cartilage degradation mediated by activating the PI3K/Akt/mTOR signaling [63]. Chen et al. found that the lncRNA SNHG15 inhibited the progression of OA through extracellular matrix homeostasis and directly regulated the miR-7/KLF4/β-catenin axis [64]. A recent study has shown that Semaphorin 6D (SEMA6D) is an inhibitor of OA and a direct target gene of miR-7, and the miR-7/SEMA6D axis can regulate the occurrence of OA by mediating the p38 pathway [65]. Taken together, miR-7 provides a potential therapeutic target for improving OA.

### 3.8. Aging

The aging of cells will lead to the disorder of the skin microenvironment, and the ability to regulate inflammation will be weakened. Fibroblasts are one of the key cells in maintaining youthful skin, and age-related defects in fibroblast differentiation and function are associated with impaired functions of hyaluronan synthase 2 (HAS2) and EGFR. This is due to the upregulation of miR-7 expression. In senescent fibroblasts, inhibition of miR-7 prevented HA-mediated dysregulation of the CD44/EGFR signaling pathway [36]. In addition, data confirmed that the STAT1 binding site was found on the miR-7 promoter. Using the anti-inflammatory steroid 17β-estradiol can activate the estrogen receptor to attenuate the expression and activity of STAT1, further downregulate miR-7 and upregulate EGFR, and restore aging fibroblasts to a “young state” [26]. Therefore, miR-7 may be a potential target to restore the ability of chronic wound healing in the elderly.

### 3.9. Inflammation Drives Malignancy

Many studies have confirmed that inflammation is a risk factor for many malignant tumors. Inflammatory mediators can induce genetic and epigenetic changes such as DNA methylation, tumor suppressor gene point mutations, and post-translational modifications, causing changes in key pathways that maintain the stability of the normal intracellular environment and leading to the occurrence and evolution of cancer [82,83]. In recent years, the study of miR-7 in inflammation has further deepened our understanding of the correlation between inflammation and tumors.

Inflammation-dependent induction of miR-7 was found in mouse gastric tumors. As a tumor suppressor, miR-7 was downregulated in mouse gastritis and gastric tumor tissues. Furthermore, the level of miR-7 was negatively correlated with the level of pro-inflammatory cytokines, suggesting that the severity of the inflammatory response was related to the downregulation of miR-7. Further studies found that the stimulation of macrophage-derived factors can cause the downregulation of miR-7, and downregulation of miR-7 expression is important for maintaining the undifferentiated state of gastric epithelial cells, thus promoting the occurrence of gastric tumors [66].

Studies have shown that chronic inflammation can trigger TLR signaling, leading to cancer. Studies have found that TLR9 can promote the progression of lung cancer [13], and miR-7 can inhibit the growth and metastasis of lung cancer cells by regulating the PIK3R3/Akt pathway and inhibiting the TLR9 signaling pathway [14]. Another more in-depth study found that TLR9 can enhance the expression of human antigen R (HuR) in human lung cancer cells through the Akt pathway. Upregulated HuR can bind to the loop site of pri-miR-7 and ultimately reduce the expression of miR-7, thereby cooperating with PI3K/Akt pathway transduction to form a positive feedback loop, which eventually leads to enhanced growth and metastatic potential of human lung cancer cells [15].

It is well known that the hepatitis B virus (HBV) is closely associated with liver cancer, and the correlation rate between the two is as high as 80%, making HBV the second most common cause of cancer after tobacco. Chen et al. found that HBV-encoded X (HBx) enhanced miR-7, -103, -107, and -21 by inducing abnormal activation and nuclear translocation of the inhibitor-κB kinase-α (IKKα). These elevated miRNAs inhibit maspin expression from promoting HCC progression and are strongly associated with poor survival in HBV-associated HCC patients [67]. Interestingly, another study found that HBx can upregulate miR-7 expression to target 3’UTR of EGFR mRNA, which in turn results in the reduction of EGFR protein expression in HCC cells. Therefore, HBx, through miR-7-mediated EGFR inhibition, causes HCC cells to exhibit slow growth behavior [68].

### 3.10. Other Inflammation-Related Diseases

Acute lung injury (ALI) is usually caused by systemic infection or lung-related inflammation. Zhao et al. found that miR-7 was upregulated in the lung tissue of LPS-induced ALI model mice and that miR-7 deficiency can improve the lung pathology of ALI mice by targeting KLF4. In addition, the total number of inflammatory cells and the level of pro-inflammatory factors in bronchoalveolar lavage were significantly reduced [69].

Low-grade chronic inflammation plays an important role in many diseases, including diabetes. The latest study confirmed that miRNA-7 is the most significantly elevated among all detected miRNAs in patients with gestational diabetes mellitus (GDM). Further studies have found that miRNA-7 can regulate MAP kinase signaling by targeting insulin receptor substrate 1/2 (IRS1/2) and RAF1, leading to chronic low-grade inflammation and promoting the progression of GDM [70].

Rotavirus (RV) is a virus that causes gastroenteritis in infants and newborn animals. Except for prophylactic attenuated live vaccines, no effective drug for treating RV is currently available. However, a recent study found that miR-7 is upregulated during RV replication, and miR-7 can inhibit RV replication in vitro and in vivo by downregulating the expression of Nonstructural protein 5 (NSP5) [71].

Patients with chronic atrophic gastritis (CAG) infected by Helicobacter pylori will develop spasmolytic polypeptide-expressing metaplasia (SPEM). Chen et al. confirmed that miR-7 expression was downregulated in SPEM lesions. Further studies have found that the Chinese medicine Yiwei Xiaoyu Granules (YWXY) can restore the expression of miR-7 by regulating TFF2 [72].

Inflammation may lead to an increased risk of cardiovascular disease in Human immunodeficiency virus 1 (HIV-1)-infected patients. Despite some treatments, HIV-1-infected individuals showed increased inflammation and microbial translocation compared with uninfected individuals. In addition, differentially expressed miRNAs were found, among which miR-7 was the most significantly upregulated miRNA. miR-7 is associated with several systemic inflammatory markers, suggesting a potential role in regulating persistent chronic inflammation in HIV-1 infection [73]. However, the exact mechanism remains to be elucidated.

The TLR-4 signaling pathway plays a key role in the innate immune inflammatory response and broadly triggers the development of various clinical diseases. In a recent study, Chen et al. found that the expression of miR-7 was significantly increased in bone marrow-derived macrophages activated by the TLR4 signaling pathway under LPS stimulation. Importantly, miR-7 deficiency significantly enhanced the production of related inflammatory cytokines, accompanied by increased TLR4 signaling. In contrast, miR-7 over-expression produced opposite results by targeting the family with sequence similarity 177, member A (FAM177A) [74].

## 4. Summary and Prospect

Emerging evidence shows that regulatory RNAs, especially miRNAs, can regulate the body’s inflammatory response in multiple ways by targeting the expression of key signal transduction molecules in inflammation. One of the important members of the miRNA family discovered in recent years is miR-7. As described herein, miR-7 is emerging as a key regulator of inflammatory responses. However, there are still some urgent scientific problems to be solved regarding the regulatory mechanism of miR-7 in the occurrence and development of inflammatory responses in the body, as well as in the research on inflammation-related diseases. It mainly includes the following three aspects.

First, we have to consider the regulatory factors controlling the expression of miR-7. In humans, the expression of miR-7 is derived from three genomic loci, namely miR-7-1 (located in the intron of the HNRNPK gene on chromosome 9), miR-7-2 (in the intergenic region of chromosome 15), and miR-7-3 (in the intron of PGSF1a on chromosome 19). Each miR-7 gene is transcribed as a unique primary transcript (pri-miR-7-1, pri-miR-7-2, and pri-miR-7-3), which is subsequently processed into three unique precursor transcripts (pre-miR-7-1, pre-miR-7-2, and pre-miR-7-3). These products are ultimately processed into the same mature miR-7 sequence of 23 nucleotides [9,10]. As mentioned above, our research group found that the precursor sequence miR-7b is the main source of mature miR-7 expression in the activation of mouse CD4^+^T cells [27]. However, it is necessary to identify the primary transcript that plays a key role in the eventual transformation into the mature miR-7 sequence in a certain cell or molecule. In addition to transcripts, numerous regulatory factors affecting the expression of miR-7 were also introduced in this paper, such as methylation, circRNAs, lncRNAs, transcription factors, and TDMD [Figure 2]. Of course, there are still some phenomena that have not been found in inflammation-related diseases, but the existence of these phenomena cannot be ruled out. For example, miR-7 is an important regulator of lung cancer, and its downregulation mechanism in lung cancer remains unclear. We found a G→C change at the -617 site (25/39, 64.1%) and an A→G change at the -604 site (20/39, 51.3%) in the miR-7 promoter region in lung cancer tissues. Further studies confirmed that mutations at these sites could reduce the activity of the miR-7 promoter and change the expression level of miR-7 [84]. This miRNA is evolutionarily conserved in most sequenced bilateral species, with three genes required to generate mature miR-7 sequence. There are multiple regulatory factors that ensure that miR-7 can be fully transcribed, indicating that miR-7 plays a core functional role in the body’s pathophysiological process.

The second aspect is the complexity of its relationship with the pathogenesis of the disease. The inflammatory process involves the participation of various inflammation-related cells and molecules, and miRNAs are differentially expressed in different inflammatory cell populations and molecules, especially immune cells and molecules [Figure 4]. Immune cells are complex, heterogeneous subpopulations that are constantly renewed in the body and can exist at different developmental stages or perform different functions simultaneously. For example, CD4^+^T cells are divided into TH1, TH2, Treg, and other subgroups. Studies have shown that miR-155 is induced when T cells are activated and promotes Th1 differentiation when over-expressed in activated CD4^+^T cells. However, CD4^+^T cells lacking miR-155 tend to be Th2 differentiated, suggesting that the loss of miR-155 may alter the differentiation of CD4^+^T cells [85]. Mφ is highly plastic, capable of dynamic transitions between M1 (pro-inflammatory) and M2 (anti-inflammatory) polarized phenotypes. Banerjee et al. found that the expression level of Let-7c in M2 is higher than that in M1, and Let-7c can play a key role in maintaining M2 activation by inhibiting the pro-inflammatory transcription factor C/EBPσ. In M1-polarized macrophages, ectopic expression of Let-7c reduced IL-12 levels and major histocompatibility complex class II surface expression, suggesting suppressed inflammatory activity [86]. The current research on miR-7 is limited to the preliminary classification of immune cells. Therefore, the exact mechanism by which miR-7 regulates different immune cell populations and by which related molecules participate in the occurrence and development of inflammation-related diseases still needs further in-depth study.

The third aspect is its research as a disease intervention target. Many studies have confirmed that miR-7 can play an important role in the biological regulation of various inflammation-related diseases by inhibiting the expression of different target molecules, reflecting the complexity of the regulatory mechanisms of different miRNA expression in different cell and molecular types [Figure 3]. Therefore, some potential problems still need to be further resolved in future clinical treatment [Figure 5]. First, current studies have shown that miR-7 can regulate multiple target molecules in different biological processes. Moreover, accumulating evidence indicates there might be complex networks among these distinct targets in inflammation-related diseases. For example, YY1, as a target of miR-7, induces the deactivation of C/EBP-α expression by decreasing C/EBP homologous protein,10 (CHOP-10) expression [38]; however, C/EBP-α can regulate the expression of miR-7 [27,56], forming a negative feedback loop [Table 1] [Figure 6]. However, the exact correlations among these targets during the process of different inflammation-related diseases remain to be further elucidated, raising the possibility of off-target side effects of miR-7 in treating diseases [87]. Second, the feasibility of targeting miRNAs to specific tissues and cells remains the key to gene therapy. For example, the corticotropin releasing hormone receptor 2 (CRHR2)/urocortin 2 signaling pathway specifically downregulates the expression of YY1 through the increase of miR-7, thereby enhancing the transcriptional activity of Fas. However, in SW620-CRHR2+ cells with high expression of miR-7 and HCT116 cells with low expression of miR-7, the regulation of miR-7 has opposite effects on the expression of YY1 and Fas as well as cell sensitivity to ch11 killing, suggesting the complexity of its development as a target [88]. Currently, many articles have reported that target gene expression can be achieved by using tissue- and cell-specific promoter manipulation [16,89]. However, the selection of specific promoters and the transformation of artificial promoters, especially the optimization and modification of promoters, still require further verification. In addition, exploring promoter-related characteristic transcription factors, promoter epigenetic changes, and their integrity will also contribute to the rapid development of promoter-targeted gene therapy research. The third is the delivery route and delivery system. Rational selection of drug delivery routes will further improve the implementation of gene therapy. For example, Babae et al. developed a novel αvβ3/αvβ5 integrin-targeted biodegradable polymeric nanoparticle that can effectively deliver miR-7 mimic systematically to both endothelial cells and tumor cells [90]. In addition, using polyethylene glycol and other delivery systems with low toxicity, low immunogenicity, a long metabolic cycle, and easy modification is of immense value for improving the efficacy of gene therapy and reducing potential side effects [89].

In conclusion, in-depth studies have been conducted on miR-7 in the inflammatory process in recent years. There is no doubt that miR-7, a key gene regulator, affects almost all aspects of the inflammatory response in pathological settings. With the increasing knowledge of miR-7 biology and the development of new methods to efficiently deliver miR-7 modifiers to specific cell subpopulations, we are confident that targeted manipulation of miR-7 pathways will soon become a viable option for treating a range of human inflammatory-related diseases.

## Figures and Tables

**Figure 1 biomolecules-13-01185-f001:**
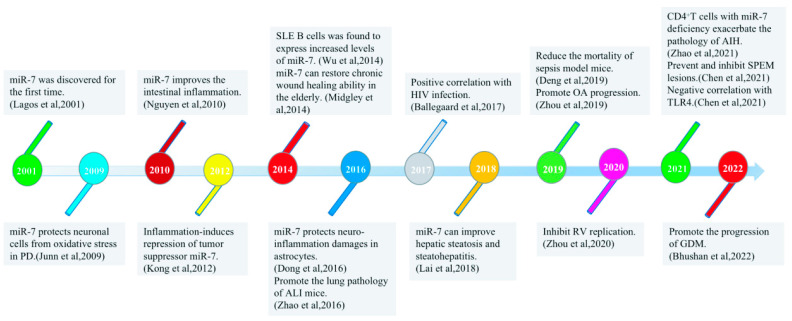
Timeline of major research discoveries on miR-7 with inflammatory-related diseases.

**Figure 2 biomolecules-13-01185-f002:**
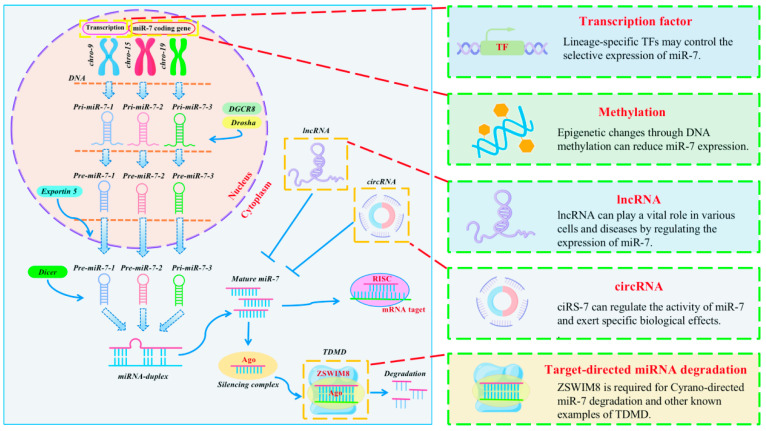
Regulatory mechanism of miR-7 expression in inflammation. MiR-7 is transcribed from three distinct genomic sites on chromosomes 9, 15, 19 into primary miR-7 transcripts (pri-miR-7-1, pri-miR-7-2, pri-miR-7-3, respectively), where transcription factors and DNA methylation influence transcription levels in inflammation. The hairpin precursor molecules (pre-miR-7-1, pre-miR-7-2, pre-miR-7-3) were subsequently processed by DGCR8 and Drosha and exported from the nucleus to the cytoplasm by Exportin 5. Then it is further transformed into the same mature miR-7 sequences by Dicer processing. In general, these sequences are incorporated into the RISC complex and guided to miR-7 target mRNAs to inhibit their expression. However, in inflammation, TDMD, LncRNA and circRNA can regulate the expression of mature miR-7, respectively.

**Figure 3 biomolecules-13-01185-f003:**
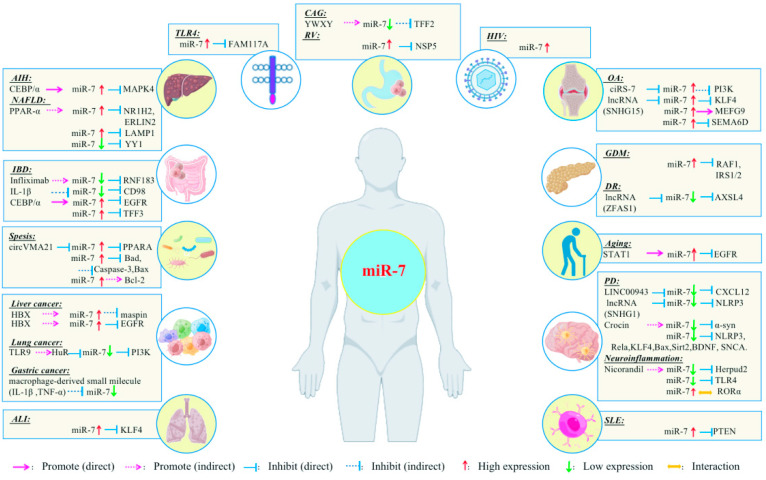
Relationship between miR-7 and inflammation-related diseases.

**Figure 4 biomolecules-13-01185-f004:**
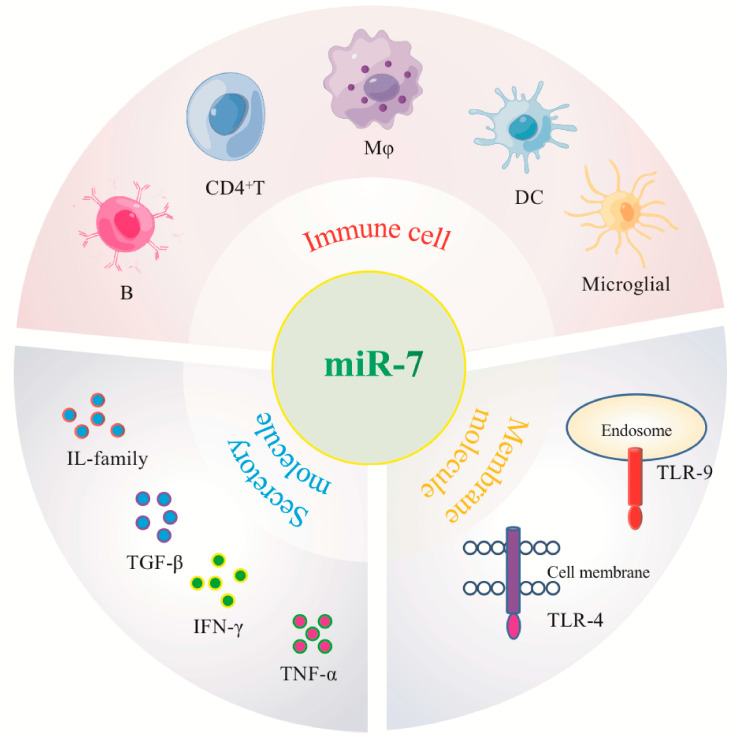
MiR-7 regulates immune cells and molecules associated with different inflammatory-related diseases.

**Figure 5 biomolecules-13-01185-f005:**
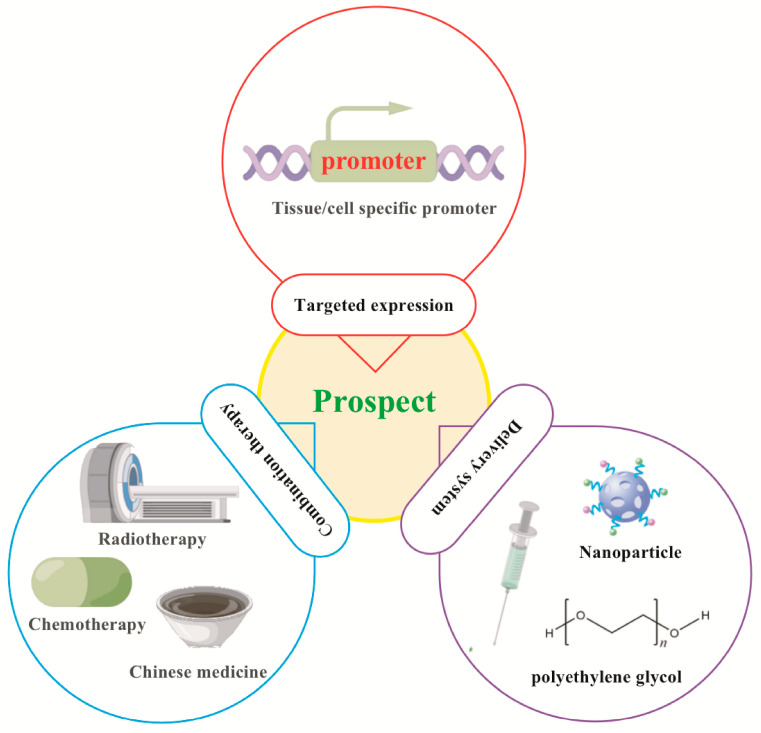
Prospects for targeted manipulation of miR-7 in the treatment of inflammation-related diseases. Targeted expression: Tissue/cell specific promoter- operating target expression of gene therapy in tissue/cell. Delivery system: long metabolic cycle, easy modification, low toxicity and low immunogenicity. Combination therapy: Combined with Chinese medicine, chemotherapy, radiotherapy.

**Figure 6 biomolecules-13-01185-f006:**
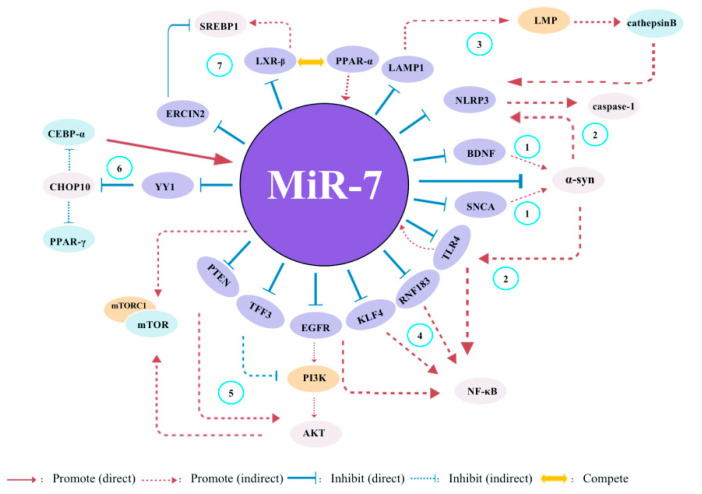
The potential global network of miR-7 target genes in inflammatory-related diseases. (**1**) miR-7 regulates the BDNF/α-syn axis in the early stages of PD. Moreover, it also can regulate α-syn expression by targeting SNCA. (**2**) In adult neural stem cells (ANSCs)s, miR-7 can control the expression of NLRP3 and α-syn, subsequently activating both TLR4/NF-κB and NLRP3/caspase-1 signals. (**3**) Meanwhile, hepatic sEVs contribute to endothelial hyperpermeability in coronary microvessels by delivering novel-miR-7 and targeting the LAMP1/LMP/Cathepsin B/NLRP3 inflammasome axis during NAFLD. (**4**) miR-7 can inhibit the expression of TLR4, RNF183, KLF4 and EGFR, thereby regulating the activaton of NF-kb pathway. (**5**) miR-7 inhibits the expression of PTEN, TFF3 and EGFR, as well as mTROC1, thereby regulating the activaton of PI3K/Akt/mTOR pathway. (**6**) YY1, as a novel target of miR-7, induces the transactivation of C/EBP-α and PPAR-γ expression by decreasing the expression of CHOP-10. Meanwhile, C/EBP-α can regulate miR-7 expression, indicating a nagative feedback loop. (**7**) PPAR-α-mediated activation of miR-7 expression further suppresses LXR signaling. While PPAR-α inhibits LXR-mediated SREBP1 transcriptional activation, stimulating miR-7 expression appears to rheostat this effect through suppression of a negative regulator of SREBP1 activity (ERLIN2).

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
