# Peer review of "MicroRNA-7: A New Intervention Target for Inflammation and Related Diseases"

_biomolecules, 2023, doi:10.3390/biom13081185_

Round 1
Reviewer 1 Report
In this paper, the authors present a review of the role of the microRNA MIR-7 in the context of inflammation and related diseases.
The manuscript is divided in several sections. The authors first write an introductory section regarding general features of the MIR-7 gene family. Then they investigate the possible molecular mechanism by which the MIR-7 precursors can be regulated at the expression level. In the third part, they concentrate on MIR-7 and inflammatory diseases discussion, with the following main sub-sections: Hepatitis, Parkinson’s disease (PD), Neuroinflammation, Inflammatory bowel disease (IBD), Sepsis, Systemic lupus erythematosus (SLE), Osteoarthritis (OA), Aging, other Inflammation-related diseases.
The major point here is the discussion of the MIR-7 non-coding RNA, that is one of the first discovered and most-studied miRNA gene family, with potentially hundreds of papers associated to it and know molecular functions in almost every biological context.
Given the complexity of the MIR-7 biology, I think that a review like the one presented by Chao Chen and colleagues, even if limited in its scope, could be useful for interested readers. Specifically, the paragraph discussing the MIR-7 expression regulation is interesting.
Overall my opinion is that the manuscript is properly and correctly written and designed.
Major:
As a general comment to the authors, I would like to point out that the function of a microRNA is fundamentally linked to the function of the set of target genes and the relative interaction network. The authors show some target genes of the MIR-7 family in Figure 3 and Table 1.
However, a description of the potential global network of MIR-7 target genes is lacking here. Even if this is a very complex topic, I would like to see at least a general discussion of the set of potentially target genes of the MIR-7 family, for example using some of the many resources available, like TargetScan and MirTarBase. In other words, I would like to see a paragraph, mainly devoted to non-expert readers in which a basic description of the potential regulatory network (in the human genome) under the control of the MIR-7 family is presented.
I do not have specific concerns on the English Language used here.
Reviewer 2 Report
Chen et.al., has extensively reviewed the role of miRNA-7 in different diseases. Along with that, they have also covered the regulation of miR-7 by different mechanism by methylation, lncRNA etc. Since, authors have extensively reviwed the miR-7 and this would be an important articel to refer, I would suggest to include whether the miR-7 is detected as circulating or packaged into Extracellular vesicles. It will be important to highlight in the table 1 whether the role of miR-7 identified is via EVs or direct traget. If this inofrmation is available for any disease listed in Table 1, it will be imperative to know the source of miR-7
Round 2
Reviewer 1 Report
This is revised version of the miR-7 review provided by Chao Chen and colleagues.
The authors added a figure (Figure 6) with some information and description of miR-7 - target gene interactions.
I think this is adeguate as response to my suggestions.